# A Multi-Epiphysiological Indicator Dog Emotion Classification System Integrating Skin and Muscle Potential Signals

**DOI:** 10.3390/ani15131984

**Published:** 2025-07-05

**Authors:** Wenqi Jia, Yanzhi Hu, Zimeng Wang, Kai Song, Boyan Huang

**Affiliations:** College of Electrical Engineering and Information, Northeast Agricultural University, Harbin 150030, China; jiawenqi@neau.edu.cn (W.J.); huyanzhi@neau.edu.cn (Y.H.); wangzimeng@neau.edu.cn (Z.W.); s231402007@neau.edu.cn (K.S.)

**Keywords:** emotional classification, non-invasive physiological, skin potential, muscle potential, XGBoost

## Abstract

Real-time emotion monitoring in pet dogs is essential for ensuring their well-being and improving interaction with humans. This study proposes a practical emotion classification system using four observable physiological signals—skin potential, muscle potential, respiration frequency, and voice pattern. Leveraging a compact, non-invasive sensor and the XGBoost algorithm, the system enables accurate real-time detection of canine emotional states, particularly abnormal ones. It offers a portable and efficient solution for everyday monitoring of dog emotions in practical settings.

## 1. Introduction

Dogs, as highly empathetic and socially integrated animals, play an increasingly significant role in human life. They serve in diverse capacities such as search and rescue, guiding the visually impaired, emotional support, and companionship, thereby becoming indispensable to modern society [1]. However, abnormal emotional states in dogs can negatively impact their own physical and psychological health, while also posing potential safety risks to the public. Incidents involving aggressive canine behavior have raised societal concerns, making it increasingly challenging for dogs to coexist freely in human environments [2]. Therefore, effective monitoring of canine emotions, particularly the detection of abnormal states, is of great importance. Real-time tracking of physiological signals combined with emotion classification enables owners to better understand their dog’s psychological status, thereby enhancing human–animal interaction. More importantly, timely identification of negative emotional responses allows owners and bystanders to take precautionary measures to mitigate harmful behaviors [3].

Real-time monitoring of dog emotional states not only holds significant implications for human society, but more critically, serves to safeguard the physical and psychological well-being of the dogs themselves, particularly in the early identification of abnormal emotions. In this study, abnormal emotional states are defined as negative affective conditions, specifically including sadness, anger, and fear. Emotional assessment can be conducted either through human observation [4] or by employing machine learning algorithms [5]. When dogs experience such negative emotional shifts, distinct physiological changes are often observed. Behaviors such as fear-induced withdrawal or anger-driven aggression are closely associated with neurophysiological and hormonal responses. For instance, exposure to fear-related stimuli has been shown to elevate heart rate, body temperature, cortisol, and progesterone levels in dogs [6]. In severe cases, acute stress responses induced by excessive fear may lead to fatal outcomes, necessitating pharmacological intervention with anxiolytic or sedative agents for emotional regulation [7]. These findings highlight the critical importance of timely emotion recognition. Moreover, prolonged sadness in dogs may signal underlying pathological conditions. Therefore, early detection of abnormal emotions can facilitate preclinical diagnosis and enable prompt medical intervention, contributing significantly to animal welfare.

However, accurately detecting and classifying dog emotions remains a significant challenge. Traditional behavioral observation methods [8] are limited in their ability to capture the true emotional states of dogs in real time and often lack objectivity. Consequently, increasing research attention has shifted toward the analysis of physiological indicators in dogs. By investigating the correlation between physiological responses and emotional changes, researchers have developed emotion recognition systems that are less susceptible to external interference and capable of real-time monitoring of a dog’s psychological state.

Dogs exhibit distinct behavioral patterns corresponding to different emotional states. Accordingly, computer vision models have been employed to analyze and interpret facial and limb movements, particularly focusing on features such as the eyes, ears, and tail. Ref. [9] introduced a facial action coding system for dogs (EMDOGFACS), which associates specific facial actions with emotional expressions. Their findings highlighted that certain basic emotions in dogs are conveyed through identifiable movements, with ear-related actions playing a key role. Subsequent studies have leveraged image processing techniques to classify dog emotions via facial and body gesture recognition, particularly by analyzing muscle-driven movements of the ears and eyes [10,11,12]. In ref. [13], researchers collected and analyzed facial features—including ear position—from 28 pet dogs to distinguish between positive and negative emotional states. The study further enabled classification based on common actions such as blinking, lowering the chin, and nose-licking, and contributed to the refinement of the DogFACS coding system. A summary of frequently observed facial and behavioral indicators used in emotion classification is presented in Table 1 [14]. From a physiological perspective, such behavioral actions are known to induce changes in MP and RF, thereby offering theoretical support for the present study’s focus on observable physiological signals.

In addition to the visual method, vocalizations have also been utilized to infer canine emotional states [15,16]. Researchers commonly apply overlapping frame techniques to smooth the extracted acoustic contours, followed by feature extraction using methods such as Principal Component Analysis (PCA) [17,18] and Latent Dirichlet Allocation (LDA) [19]. LDA has achieved an emotion recognition accuracy of 69.81% in dog vocalization analysis [20]. A variety of machine learning algorithms have been adopted to train speech-based emotion recognition models, including Hidden Markov Models (HMM) [21], Gaussian Mixture Models (GMM) [22], Artificial Neural Networks (ANN) [23], and Support Vector Machines (SVM) [24,25]. These approaches typically yield classification accuracies around 70%, according to ref. [26], as summarized in Table 2.

Notably, ref. [27] applied a novel machine learning algorithm to analyze dog barks, incorporating context-specific and individual-specific acoustic features, and achieved a classification accuracy of 52%. More recently, ref. [28] proposed a multi-hop attention model for speech emotion recognition, which combined a BiLSTM network for extracting latent features from vocal signals with a multi-hop attention mechanism to refine classification weights. This approach significantly improved recognition performance, reaching an accuracy of 88.51%.

Currently, dog emotion classification primarily relies on visual processing and speech recognition techniques. However, visual methods become increasingly challenging in scenarios involving multiple dogs, as overlapping subjects and complex backgrounds hinder effective image segmentation. Moreover, the exponential growth in convolutional layer settings increases computational complexity and reduces the efficiency of edge feature extraction. In addition, image-based emotion recognition lacks routine applicability in real-world environments. Similarly, in multi-dog environments, overlapping vocalizations introduce significant interference, making it difficult to isolate the vocal signals of a specific dog.

To overcome the aforementioned limitations, we propose a multi-modal approach to emotion classification that moves beyond reliance on a single indicator. Specifically, we utilize vocal signals, which are both highly accurate and well suited for wearable device integration, in combination with muscle potential (MP) signals—associated with behavioral responses—and skin potential (SP) signals—reflecting neuroelectrical activity. Respiration frequency (RF) is additionally employed as a corrective parameter. Together, these modalities enable a comprehensive and real-time assessment of canine emotional states.

This model enables dog owners to assess whether their pet is in a normal emotional state. When a dog experiences abnormal emotions—such as anger or fear, as discussed in this study—it may exhibit behaviors that are detrimental to its own physical and psychological well-being, as well as to the safety of surrounding humans. For example, a dog that feels angry, tense, or on alert in response to environmental stimuli may display aggressive behavior toward passersby. Similarly, excessive fear can trigger severe stress responses, including acute panic, which in extreme cases may lead to cardiac arrest and death. Timely emotion monitoring using this system allows breeders or owners to take early corrective actions, thereby safeguarding the dog’s health and preventing potential harm to others. Moreover, the system may also assist in the early detection of disease. Emotional anomalies such as persistent unhappiness may indicate underlying physical pain or the early onset of illness. In summary, accurate emotion classification enabled by a lightweight, wearable device offers substantial benefits to dogs, their caretakers, and the surrounding environment.

A wearable emotion classification system for dogs not only enhances the management of individual animal welfare, but also drives innovation and sustainable development in the broader field of animal science. Overall, this paper makes the following contributions:

(a) An integrated emotion classification framework based on four observable physiological signals (SP, MP, RF, and VP) is introduced, and four machine learning algorithms were trained and evaluated. The results demonstrate the superiority of the multi-modal signal fusion in improving classification accuracy and confirm the global optimality of the selected algorithm—XGBoost.

(b) The framework introduces the application of two types of electrophysiological signals—skin potential (SP), associated with neurophysiological activity, and muscle potential (MP), associated with behavioral responses—for canine emotion classification. These signals were measured and analyzed across four dog breeds (including one small, one medium, and two large breeds), enabling a comparative study of signal characteristics across breed types.

(c) SHAP (Shapley Additive exPlanations) analysis was applied to rank the relative contributions of the four physiological indicators to emotion classification, as well as to assess the importance of specific features within each modality. This interpretability analysis offers valuable insights for the development of future wearable emotion recognition systems for animals.

The remainder of this paper is organized as follows: Section 2 formulates the signal acquisition methods and model architecture. Section 3 shows the experiments of the proposed multi-epiphysiological indicator dog emotion classification system framework. Comparison and discussions are presented in Section 4. Finally, Section 5 concludes the entire paper.

## 2. Materials and Methods

### 2.1. Signal Acquisition and Feature Extraction

The complete epiphysiological monitoring system proposed in this study is fully non-invasive. All four physiological signals—SP, MP, RF, and VP—were acquired using surface-attached sensors placed at anatomically relevant locations. In compliance with the ethical principles of Replacement, Reduction, and Refinement (3Rs) [29], we designed a controlled emotion-elicitation experiment to capture physiological responses associated with distinct emotional states.

To minimize environmental confounders, such as ambient noise, temperature variation, and lighting fluctuations, the experiments were conducted in a controlled environment with the following conditions: a quiet setting, ambient temperature maintained at 26 °C ± 1 °C, uniform lighting with a lighting coefficient of 1:10. Given the system’s intended application in real-life pet scenarios, we prioritized naturalistic and ecologically valid emotion elicitation. Emotion-inducing stimuli were selected from commonly used pet-interaction products and daily social situations. All participating subjects were domesticated companion dogs, voluntarily recruited from two local pet parks. Owners who expressed willingness to participate visited the laboratory in advance to verify the experimental safety and feasibility.

Considering the high individual variability in canine emotional expression, each dog’s owner—being most familiar with their pet’s behavior—was responsible for guiding the emotional transitions. This approach ensures ecological validity and aligns with potential real-world applications. For instance, emotional states were induced through personalized scenarios. For example, using dog treats to induce the dog’s happy emotion, some dogs will induce the sad emotion when their owner refuses their request to go out to play, some dog will induce anger when protecting food, and some dogs will induce fear when they hear the voices of more ferocious animals.

The emotion-elicitation approach used to induce specific emotional states in pet dogs has proven to be highly effective and aligns well with established principles in animal behavioral science. In dogs, for example, reward-related brain activation is commonly triggered by positive stimuli such as food, verbal praise, play, and familiar human scents, all of which reliably evoke feelings of happiness [30,31,32]. Conversely, anger is typically induced by threatening stimuli, including unfamiliar barking dogs, approaching strangers, or other perceived dangers [33]. Behavioral fear-induction tests in dogs often involve sudden loud noises or novel, unexpected objects [34]. A substantial body of literature has demonstrated strong correlations between specific behavioral responses and underlying emotional states in dogs, as also summarized in Table 1 of this study.

To further enhance the real-world applicability of our research, we conducted a behavior-based emotion conversion experiment with five Border Collies trained to use pet communication buttons. These dogs were able to express their emotional intent through button presses, enabling two-way interaction with their owners. The owners monitored the dogs via real-time video surveillance and used the button interface to engage in communication when they perceived a shift in the dog’s emotional state. The moment of emotion transition was then recorded based on the timestamp of the interaction. Experimental staff cross-referenced these timestamps with physiological data to obtain corresponding values for the four monitored physiological indicators. This voice-button-based method of confirming canine emotions has been shown to be highly reliable [35], allowing for more precise labeling of emotional categories at specific moments in time.

Table 3 shows the hardware materials used in the measurements during our experiments. Throughout the recording monitoring, data loggers continuously recorded all four physiological signals during each emotional episode. Behavioral cues were used to annotate and verify the emotional state, and data entries included: (a) the emotional category (happiness, sadness, anger, fear); (b) the perceived emotional intensity (rated on a scale from 1 to 5 based on behavioral duration and severity); (c) the exact onset time and duration of each emotional episode.

To ensure accurate and stable acquisition of epiphysiological signals, two anatomical regions were selected for sensor placement based on physiological relevance and signal fidelity. For SP signal and MP signal measurements, the forelimb was chosen as the recording site due to its abundant sweat gland distribution and relatively low subcutaneous fat content, which enhance the sensitivity and stability of electrical signal detection.

For VP and RF monitoring, the neck region was selected. To collect vocal signals while minimizing interference from environmental noise and overlapping vocalizations from other animals, we adopted a non-invasive air-conduction microphone. This sensor captures acoustic signals transmitted through the air from the dog’s throat region, providing a practical solution for naturalistic emotional monitoring in daily pet care. In addition, the proximity of the laryngeal area allows for the integration of both the air-conduction microphone and a respiration sensor, enabling synchronized acquisition of vocal and respiratory features. The combined sensor unit was affixed securely around the neck to ensure continuous, high-fidelity signal collection with minimal motion artifacts. The specific signal acquisition sites are shown in Figure 1, and the actual wearing of the testing equipment is shown in Figure 2.

Due to the differences in the amplitude of the EP, MP, RF, and VP signals in different individuals, we needed to normalize all signal data. This formula normalizes the signal to the [0, 1] interval: (1)$$X_{normal}=\frac{X-X_{min}}{X_{max}-X_{min}}$$
where $X_{max}$ and $X_{min}$ represents the minimum and maximum values of the original signal.

Due to the extremely low probability of packet loss or corruption during Bluetooth transmission (typically less than 1% in practical scenarios), the SP and MP signals may occasionally fail to strictly meet the intended sampling frequency of 1000 Hz. To ensure signal integrity and uniform sampling, missing data points were reconstructed using MATLAB’s (MATLAB R2023a) built-in interp1 function. Given the rarity of packet loss, standard interpolation methods suffice to accurately restore the signal without introducing distortion. In this study, cubic spline interpolation was employed by specifying the ‘spline’ option within the interp1 function, enabling precise reconstruction while preserving the original waveform characteristics.

In total, we collected 523 emotional data samples from 30 dogs, including 13 Border Collies, 5 Samoyeds, 8 Golden Retrievers, and 4 Shih Tzus. Each sample comprises SP, MP, RF, and VP signal recordings during a 15-second period that represents a typical emotional state. After that, we mark the corresponding labels to form a dataset to build a sentiment recognition model. In this article, we use the discrete sentiment model for emotion classification. We explore only the four basic emotions (happiness, sadness, anger, and fear). Therefore, the samples include 187 happy samples, 115 sad samples, 103 anger samples, and 118 fear samples. The training set and the test set of data do not intersect, which are independent datasets, respectively.

We calculated and extracted 29 features from each sentiment sample, including 15 time-domain features, 13 frequency-domain features, and 1 nonlinear feature. Table 4 lists the names of the specific data features [36].

The extracted features include time-domain, frequency-domain, and nonlinear feature characteristics derived from the emotion-related physiological signal sequences. In the time domain, the following statistical features were computed from each emotional signal sample: the first quartile (q1), median (median), third quartile (q3), minimum ratio (min_ratio) and maximum ratio (max_ratio). To capture the dynamic variation of the signal, we also computed the first-order and second-order differentials of the signal sequence. The mean values of these differential sequences—denoted as $diff1_{m}ean$ and $diff2_{m}ean$—were used as additional time-domain features.

For frequency-domain analysis, we applied the Fast Fourier Transform (FFT) to each signal sequence to obtain its unilateral spectral components. From these, we extracted the mean frequency component ($mean_{f}$), the median frequency component ($median_{f}$), and the mean values of the first- and second-order differentials of the spectral sequence, labeled as $diff1_{m}ean_{f}$ and $diff2_{m}ean_{f}$, respectively. Among them, the minimum ratio ($min_{r}atio$) of the emotional signal sequence is calculated as follows:$$min\_ratio=\frac{X\_min}{len\_X}$$
where len_x is the data length of the signal, and x_min is the minimum value of the emotion sample sequence. The maximum ratio (max_ratio) is calculated in the same way.

The only nonlinear feature we extracted is the mean crossing rate (mcr) of the emotion sample sequence. It is based on the mean of the emotion sample sequence, and the number of times the signal passes through the mean. In other words, it is the number of times the signal changes from greater than the mean to less than the mean or from less than the mean to greater than the mean. This value represents the vibration level of the signal.

### 2.2. Classifier Model: XGBoost

We use four different machine learning algorithms based on signal features for training. Among them, the most important training model is the eXtreme Gradient Boosting (XGBoost) model. The basic steps of the XGBoost model are shown in Figure 3 below.

The XGBoosting algorithm can be used for both regression and classification. In this experiment, we use the classification function. We first give *N* training samples ${\left\{X_{i},Y_{i}\right\}}^{N}$, where $X_{i}=\left(x_{i1},x_{i2},\ldots ,x_{i14}\right)$ represents the four physiological indicator sequences of the *i*th emotion sample. $Y_{i}=\left(y_{i1},y_{i2},\ldots ,y_{i14}\right)$, is the emotional label of $X_{i}$, which marks the emotional state corresponding to the sequence of physiological indicators of the group. When the training sample belongs to the *k*th type of emotion, the vector $y_{ik}$ in $Y_{i}$ is 1, and the rest is 0. The training process of the classification model based on the XGBoost algorithm is as follows [37]:

(A) Initialize the model to train a decision tree for each class of sample *X*: (2)$$F_{k}\left(X\right)=0,\;k=1,2,\ldots ,K$$
where *K* represents the number of emotion types, and its value is 4.

(B) Construct *K* functions ${\left\{F_{k}\left(X\right)\right\}}_{k=1}^{K}$ by integrating *T* decision trees. The probability that the training sample belongs to each category is expressed as: (3)$$p_{k}\left(X\right)=\frac{exp(F_{k}\left(X\right))}{\sum_{j=1}^{K}exp\left(F_{j}\left(X\right)\right)}$$
where $F_{k}\left(X\right)=\sum_{t=1}^{T}\eta_{t}\cdot h_{t,\hat{k}}\left(X\right)$, $h_{t,\hat{k}}$ denotes the decision tree generated for category *k* in the *t*-round and $\eta_{t}$ is the learning rate.

(C) The objective function consists of a loss function and a regularization term: (4)$$F\left(X\right)=\sum_{i=1}^{N}l\left(y_{i},p\left(X_{i}\right)\right)+\sum_{t=1}^{T}\sum_{k=1}^{K}\Omega \left(h_{t,k}\right)$$

Cross-entropy loss: (5)$$l(y_{i},p\left(X_{i}\right))=-\sum_{k=1}^{K}y_{i,k}\log p_{k}\left(X_{i}\right)$$
where $y_{i,k}$ is the real label of *k*.

Regularization: (6)$$\Omega \left(h_{t,k}\right)=\gamma T_{t,k}+\frac{1}{2}\lambda ||w_{t,k}{\left|\right|}^{2}$$
where $T_{t,k}$ is the number of leaf sub-nodes of the *t*-round category *k*, $w_{t,k}$ is the leaf weight vector, and γ and λ are the regularization coefficients.

The second-order approximate expansion of the objective function is carried out, and the first-order gradient $g_{i,k}$ and the second-order Haysen matrix $h_{i,k}$ are introduced: (7)$$F_{k}{\left(X\right)}^{\left(t\right)}\approx \sum_{i=1}^{N}\sum_{k=1}^{K}\left[g_{i,k}F_{k}^{\left(t\right)}\left(X_{i}\right)+\frac{1}{2}h_{i,k}{\left(F_{k}^{\left(t\right)}\left(X_{i}\right)\right)}^{2}\right]+\Omega \left(h_{t,k}\right)$$
where: (8)$$g_{i,k}=\frac{\partial l(y_{i},p\left(X_{i}\right))}{\partial F_{k}^{\left(t-1\right)}\left(X_{i}\right)}=p_{k}^{\left(t-1\right)}\left(X_{i}\right)-y_{i,k}$$(9)$$h_{i,k}=\frac{\partial^{2}l\left(y_{i},p\left(X_{i}\right)\right)}{\partial {\left(F_{k}^{\left(t-1\right)}\left(X_{i}\right)\right)}^{2}}=p_{k}^{\left(t-1\right)}\left(X_{i}\right)\left(1-p_{k}^{\left(t-1\right)}\left(X_{i}\right)\right)$$

(D) For each candidate feature and splitting point, the gain is calculated to select the optimal split: (10)$$Gain=\frac{1}{2}\left[\frac{{\left(\sum_{i\in I_{L}}g_{i,k}\right)}^{2}}{\sum_{i\in I_{L}}h_{i,k}+\lambda }+\frac{{\left(\sum_{i\in I_{R}}g_{i,k}\right)}^{2}}{\sum_{i\in I_{R}}h_{i,k}+\lambda }-\frac{{\left(\sum_{i\in I}g_{i,k}\right)}^{2}}{\sum_{i\in I}h_{i,k}+\lambda }\right]-\gamma$$
where $I_{L}$ and $I_{R}$ are the sample sets of the left and right sub-nodes after splitting, respectively. The optimal weight of the leaf node *j* is: (11)$$w_{j,k}^{*}=-\frac{\sum_{i\in I_{j}}g_{i,k}}{\sum_{i\in I_{j}}h_{i,k}+\lambda }$$

(E) The classification model consists of a set of *K* trees ${\left\{F_{k}\right\}}_{k=1}^{K}$; each of $F_{k}=\left\{F_{k,1},F_{k,2},\ldots ,F_{k,M}\right\}$ corresponds to a prediction function for class *k*. The output probability of the model is: (12)$$P_{k}\left(X\right)=\frac{\exp(\sum_{t=1}^{T}\eta_{t}F_{k,t}\left(X\right))}{\sum_{j=1}^{K}\exp\left(\sum_{t=1}^{T}\eta_{t}F_{j,t}\left(X\right)\right)}$$
where $\eta_{t}$ is the learning rate of the *t*th tree.

For any sample $x\in R^{N}$ and category *k*, the SHAP value $\phi_{i,k}$ of eigenvalue $x_{i}$ is expressed as: (13)$$\sum_{i=1}^{N}\phi_{i,k}=f_{k}\left(X\right)-E\left[f_{k}\left(X\right)\right]$$
where $f_{k}\left(X\right)=\sum_{t=1}^{T}\eta_{t}T_{k,t}\left(X\right)$, $E[f_{k}\left(X\right)]$ is the baseline expected value.

SHAP is an interpretive method based on game theory. It assigns the impact of each feature in the prediction by calculating the contribution of each feature to the prediction. Therefore, through the analysis of the contribution values of 29 eigenvalues of each of the four physiological indicators, we can know the weight of the influence of each physiological index on the emotional changes.

The training process for a classification model is described above. Finally, the iterative trained model can use the test samples in the test set to verify the recognition effect of the model. The test process uses the Equation (Equation 12) to calculate the probability that test sample X belongs to each category. The category with the highest probability is the one predicted by the model. In addition, the algorithm related flow diagram of the entire experiment is shown in Figure 4.

## 3. Results

Through the emotion-evoked experiment, we access the data of four epiphysiological indicators. Among them, we list eight sets of typical data as shown in Table 5 below.

We present some of the experimental data in Table 6. In this study, we innovatively introduce SP and MP signals as physiological indicators for canine emotion classification. To ensure diversity and generalizability, we selected four breeds—Shih Tzu (small), Border Collie (medium), and Golden Retriever and Samoyed (large)—and analyzed their SP and MP signal patterns under four distinct emotional states: happiness, sadness, anger, and fear. The results in Table 7 show that SP signals in small dogs tend to exhibit more pronounced fluctuations, suggesting a higher degree of emotional sensitivity and reactivity [38], while medium and large dogs display relatively consistent and stable signal patterns. Interestingly, SP waveforms vary noticeably across emotional states: happiness typically presents as a gradually ascending waveform, reflecting emotional buildup; sadness manifests as a downward trend, occasionally accompanied by smaller declining waves; anger produces intense, sharply rising waveforms with large amplitudes; and fear is characterized by complex, oscillatory patterns. These distinct waveform features across emotional categories indicate that SP signals, in particular, offer strong discriminative power for emotion classification. By capturing neurophysiological SP signal, the proposed approach enhances the interpretability and precision of emotion recognition in companion dogs.

Regarding the MP signal of Table 8, we can see that there is no significant difference between body size and MP signal change. We think this is because while small dogs are more prone to mood changes, medium and large dogs have more developed calf muscles and a more complex structure, so the difference is not very obvious. The MP signal in the happy mood vibrates very intensively, showing a dense, moderate amplitude vibration. The sad emotion is similar to the MP signal state in the calm state, showing only a small amplitude and a short period of vibration. This is because dogs mostly do not produce corresponding behaviors when they are in a sad mood. The MP signal changes significantly under the angry mood, which is dense and high-level vibration. The average maximum signal value is greater than 2000, and the signal characteristics are very obvious. The vibrations are also more intense under fear, but the amplitude of the vibrations is not high, and they are not continuous vibrations of the same degree. It can be seen that although there are some differences in MP signals under different emotions, the differences are not particularly obvious.

Comparing Table 7 and Table 8, the difference in skin resistance under different emotions is obvious, while the difference in muscle resistance is not easy to distinguish. This is due to the fact that the degree of sympathetic activation in dogs varies depending on the emotion, leading to a different degree of increased sweat gland secretion [39]. This results in a significant difference in the galvanic skin responses. However, under different emotions, dogs may not act accordingly, resulting in inaccessible changes in muscle resistance. For example, when a dog is happy, it sometimes stomps its feet and produce changes in muscle resistance, but sometimes only exhibits mental activity, which does not produce corresponding actions to produce changes in muscle resistance. Although there are differences in muscle resistance when dogs stomp happily and thump on the ground angrily, the behavior in the same psychological states is also different, and implies a comprehensive judgment based on the current environment. Therefore, the difference in SP is more pronounced than muscle conductance in classifying emotions.

All classification model algorithms are implemented based on Python (PyCharm 2024.1). In this experiment, we use the training set to train a classifier, and use a five-fold cross-validation training [40] method to select the optimal parameter value from the discrete range. Meanwhile, we use the loss and accuracy for the assessment of performance. In Figure 5, we can obtain the accuracy and loss of the model from 0–100 iterations. The graph shows a noticeable reduction in fitting errors, with the suggested model proving effective after about 90 iterations.

Meanwhile, according to Figure 6, which shows the contribution rate of each feature to the classification results through the SHAP value, comparing 4 × 29 eigenvalues, we can see that SP and VP have a greater influence on the classification and judgment of emotions, and the first-order differential standard deviation (SP_diff1_std) contribution of time-domain features in the SP signal is higher than that of 4 × 29 eigenvalues. MP varies depending on the mood, but sometimes the difference is not very significant. Changes in RF can generally be used to identify abnormal emotions, but they have little effect on the classification of abnormal emotions. From this, it can be concluded that the four indicators of the SP signal, MP signal, RF signal, and VP signal have an important role in assessing whether abnormal emotions occur. However, the SP and VP signals play an important role in the judgment of mood classification.

Figure 7 illustrates the learning curves of the XGBoost algorithm in classifying four emotional states—happiness, sadness, anger, and fear—based on varying sizes of training and test sets. Each subfigure represents the algorithm’s classification accuracy under different data volumes for a specific emotion.

In Figure 7a, which presents the learning curve for the happiness category, the training set includes a relatively large number of samples (*n* = 144), resulting in a stable and well-shaped curve. Although slight oscillations are observed in the early phase of the test curve, both training and test accuracies increase steadily, indicating the model’s robustness. The final accuracy of the training set marginally surpasses that of the test set, suggesting good generalization. This further implies that classification performance can be enhanced with an increased volume of happiness-related data. In Figure 7b, the curve corresponding to sadness shows signs of underfitting, evidenced by a noticeable gap between the training and test accuracies. Since regularization was already applied during the feature extraction stage, we attribute this underfitting primarily to the limited sample size. Nonetheless, the continuous upward trend of both curves reflects the model’s learning capacity and its effectiveness in identifying sadness-related patterns. Figure 7c demonstrates an ideal learning process for the anger emotion. The curve progresses smoothly without oscillations, and both training and test accuracies improve gradually. This performance is likely due to the pronounced and consistent physiological signal variations observed in dogs under angry states. Moreover, the extraction of 29 features from each physiological indicator contributes to the model’s ability to capture discriminative patterns, highlighting the reliability of feature engineering in this context. In Figure 7d, representing the fear emotion, slight overfitting is observed as the test accuracy slightly exceeds that of the training set. However, given the small dataset size and the relatively minor deviation, the overfitting is considered negligible. The overall curve remains stable and gradually ascends, demonstrating the algorithm’s consistency.

In summary, the learning curves across all four emotions are generally smooth and exhibit an upward trend with minimal oscillation, reflecting the overall stability and robustness of the XGBoost algorithm in emotion classification. Notably, the distinct signal characteristics associated with abnormal emotions—such as anger and fear—significantly contribute to classification accuracy. These results validate the feasibility of using a combination of four physiological indicators for real-time emotional state assessment in dogs and further confirm the suitability of XGBoost for animal emotion recognition tasks.

## 4. Discussion

In this experiment, we use four different machine learning algorithms based on signal features to train the classification model. These algorithms are Neural Network (NN), Support Vector Machine (SVM), Gradient Boosting Decision Tree (GBDT), and eXtreme Gradient Boosting (XGBoost).

The most basic machine learning algorithm, NN [41], has a high accuracy rate, which can confirm the feasibility and accuracy of the four-factor comprehensive judgment of emotion categories to the greatest extent. SVM and its extension algorithm are widely used to judge emotional states using sound as a single indicator [42,43], and we also use sound as an appearance indicator in this algorithm, so it has comparative significance. It can be proved that the integration of the other three physiological indicators can better correct the accuracy of the algorithm, which is very important. GBDT [44] was used to determine sentiment categories using the SP metrics and proved to perform the best among nine machine learning algorithms [36]. Therefore, we increased the order and chose to optimize using the XGBoost algorithm to see if the results were as we thought.

In order to facilitate comparison, the multi-epiphysiological indicator dog emotion classification model based on the XGBoost algorithm is called PRO for short. All classification model algorithms are implemented based on Python sklearn. In order to achieve a higher resolution, we set some of the more important hyperparameter values in the algorithm. Table 9, Table 10, Table 11 and Table 12 lists the hyperparameter settings for each classifier model. Parameters not listed in the table use the default parameters in the sklearn library.

Figure 8 shows the accuracy of the four classification models in the face of four emotion tests. As can be seen from the images, the accuracy of each classifier model in judging different emotions is greater than 75%. It shows that the approach is very feasible and effective for judging the type of emotion by using four physiological indicators: SP, MP, RF, and VP. The accuracy of negative emotions is higher than that of positive emotions. It may be that when the dog is angry or fearful, it will make a low growl sound that is different from the usual barking, and make violent movements, so the physiological indicators are significantly different. However, the physiological indicators of happiness are mainly focused on the changes in skin electrodermal and respiratory rate, and the judgment criteria are reduced.

Figure 9 illustrates the confusion matrix case for GBDT and XGBoost. Through the display of the confusion matrix, we can compare the accuracy of the two algorithms more clearly. For example, for the 28 test samples of sad emotions, the GBDT algorithm misjudged 3 as happy emotions and 1 as fearful emotions, while the XGBoost algorithm only misjudged 3 as happy emotions and did not misjudge fearful emotions. It can be seen that the XGBoost algorithm has higher classification accuracy.

Figure 10 illustrates the ROC curves of the NN, SVM, GBDT, XGBoost(PRO) classifier models. By comparison, we can see that the curve of NN is relatively fluctuating, reflecting the probabilistic output characteristics of the neural network. The judgment of negative predictive value and positive predictive value is not very stable in sentiment classification, and it is necessary to improve the stability of the algorithm by integrating other algorithms. The curve of the SVM rises in a stepped manner because it is a good way to distinguish between the different categories. However, due to the uneven distribution of data samples and the insufficient amount of data in the test set, there is a gentle gradient rise. The curve of GBDT shows a relatively smooth exponential increase, but it is not as smooth as XGBoost, indicating that tree judgment does have a great advantage in classification calculation. The second-order calculation of XGBoost further improves the algorithm’s ability to effectively distinguish between positive and negative classes, so the corresponding AUC value is higher and the curve is smoother.

Table 13 shows in detail the relevant performance parameters of the four comparison algorithms. Comparing the four algorithms, we can see that the XGBoost algorithm is more stable and effective. Although the GBDT model is more effective in judging happy emotions, it is not as accurate in judging negative emotions as XGBoost. NN and RF, as the basic algorithms of two deep learning algorithms, are slightly lacking in classification accuracy. As an improved model of GBDT, the XGBoost algorithm is indeed more accurate in emotion classification. XGBoost accelerates convergence through the second derivative, and the regularization term reduces overfitting, which improves the classification accuracy.

## 5. Conclusions

In this study, we design a multi-epiphysiological indicator dog emotion classification system based on the XGBoost algorithm. The following three aspects of the research have been realized:

1. We demonstrate the feasibility of using SP, MP, RF, and VP to comprehensively judge animal emotion classification compared with traditional single indicator methods, achieving an accuracy of more than 75% among the mentioned machine learning algorithms.

2. The experimental results show that XGBoost outperforms NN, SVM, and GBDT in emotion classification of dogs, achieving an average accuracy of 90.54%, especially excelling in identifying abnormal emotional states.

3. We propose the novel use of SP and MP for animal emotion classification and identify SP and VP as the most influential indicators among the four, providing valuable direction for future animal emotion detection systems.

Through the combination of deep learning algorithms and multi-indicator comprehensive judgment, the proposed system is proved to produce more accurate emotional classification judgment and a faster emotional classification process. Future work will collect data on more animal species and increase the variety of classified emotions, enabling a more integrated portable animal emotion classification system.

## Figures and Tables

**Figure 1 animals-15-01984-f001:**
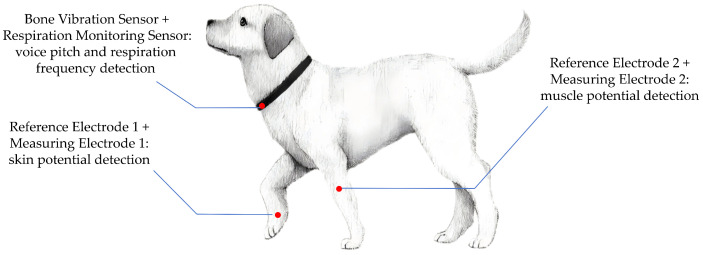
Signal acquisition sites in dogs.

**Figure 2 animals-15-01984-f002:**
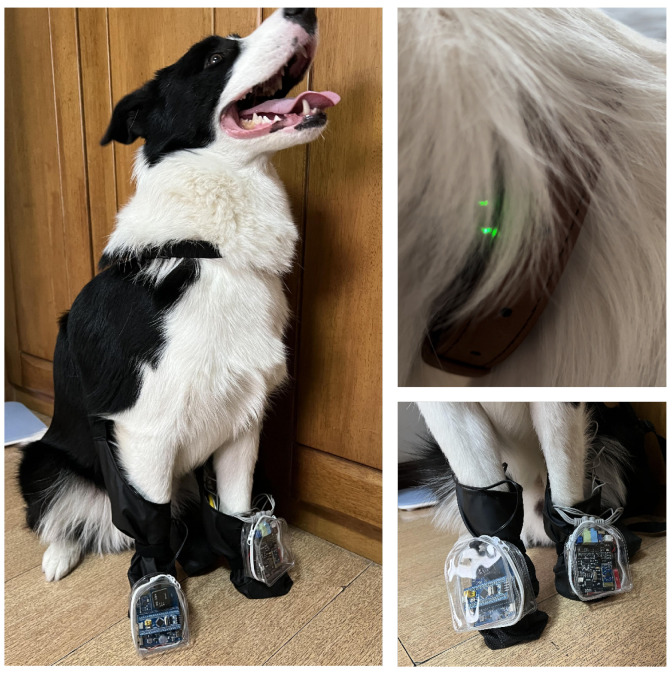
A physical device containing the signal detection instrument.

**Figure 3 animals-15-01984-f003:**
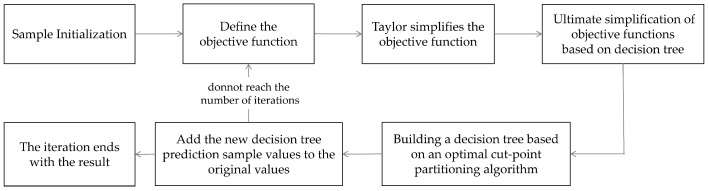
The basic steps of the XGBoost model.

**Figure 4 animals-15-01984-f004:**
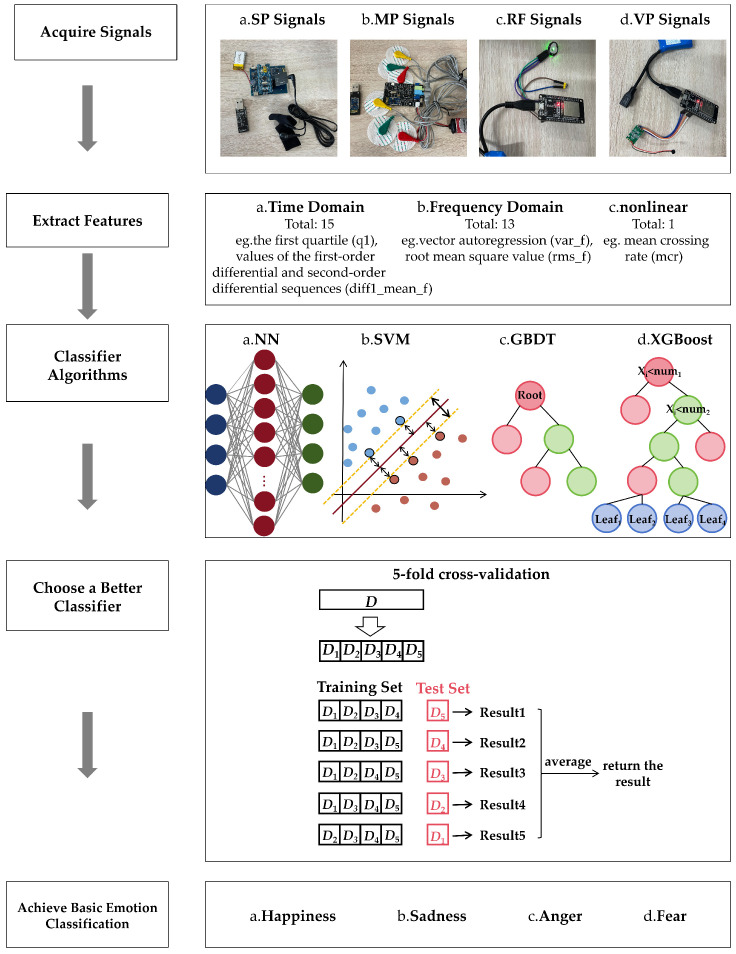
Algorithm flow chart of the four-indicator emotion classification system.

**Figure 5 animals-15-01984-f005:**
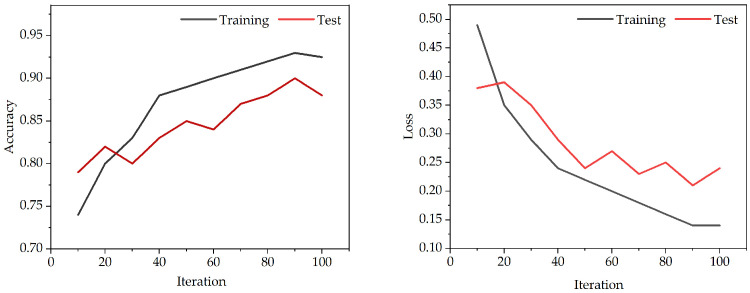
Accuracy and loss performance in the training and testing phases from 0–100 iterations.

**Figure 6 animals-15-01984-f006:**
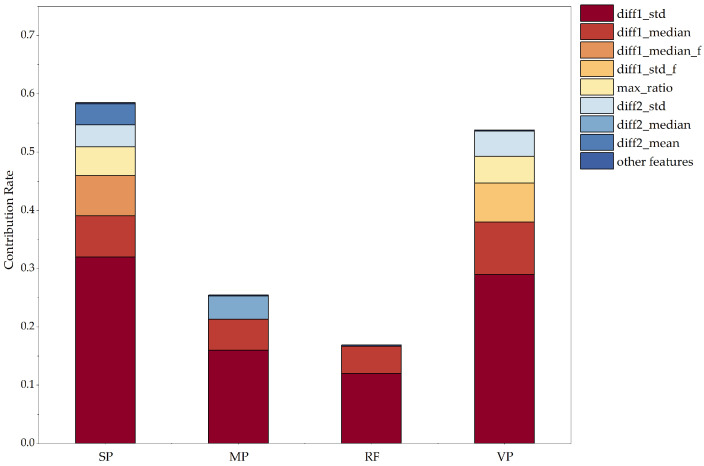
The contribution rate of four physiological indicators.

**Figure 7 animals-15-01984-f007:**
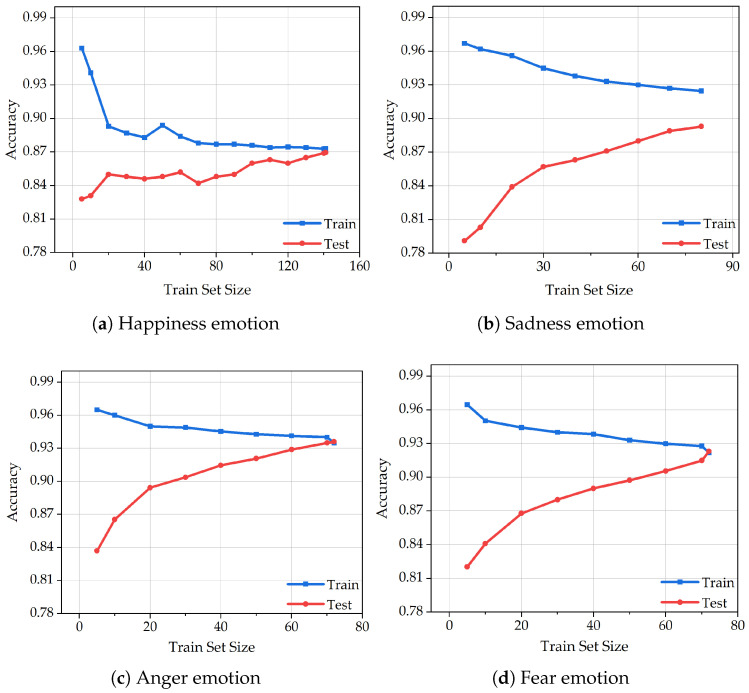
The training set size learning curve of the XGBoost algorithm under four emotions.

**Figure 8 animals-15-01984-f008:**
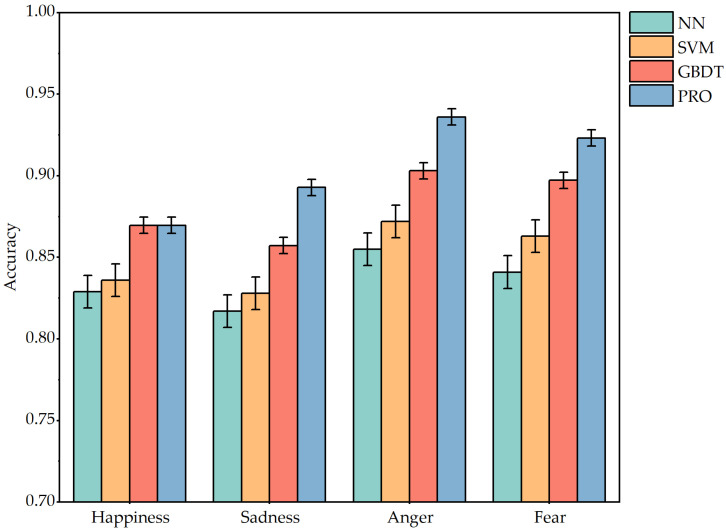
The accuracy of the NN, SVM, GBDT, and **PRO** classifier models for four emotions.

**Figure 9 animals-15-01984-f009:**
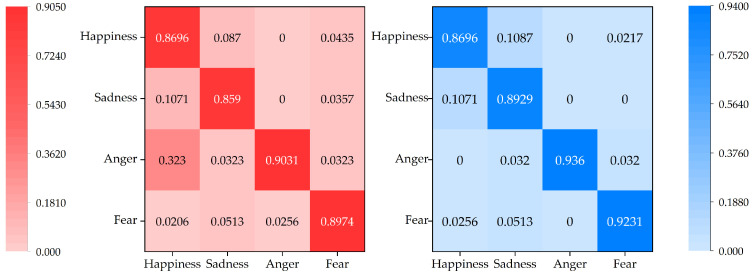
Confusion matrix plots of the GBDT (**left**) and **PRO** (**right**) algorithms.

**Figure 10 animals-15-01984-f010:**
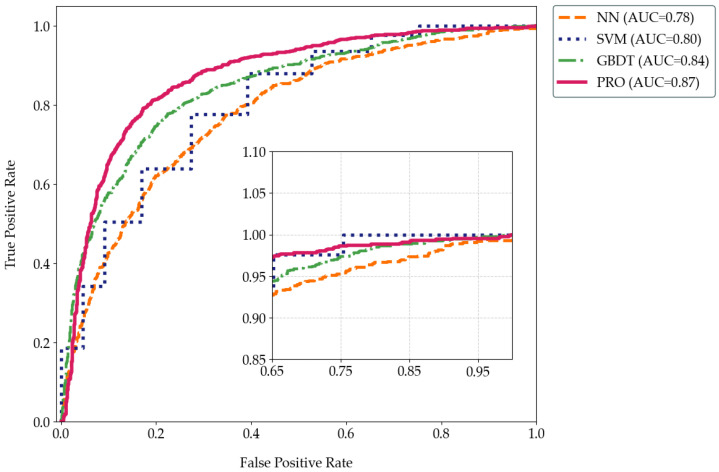
ROC comparison of NN, SVM, GBDT, **PRO**.

**Table 1 animals-15-01984-t001:** Correspondence between apparent morphology changes and emotions in dogs.

Indicators Inferring Emotions	Emotions/Affective States
Ears sway back and forth from side to side	Happiness
Ears forward and slightly cocked	Happiness
Bristle the tail	Happiness
Ears back	Sadness
Ears back and flutter slightly	Anger
Bristle the coat	Fear/Anger
Tuck their tail between the legs	Fear
Roll their ears back	Fear (alertness)
Ears forward	Fear (alertness)

**Table 2 animals-15-01984-t002:** The accuracy of four machine learning algorithms in speech recognition emotion experiments [21].

Classifier	HMM	GMM	ANN	SVM
Average	75.5–78.5%	74.83–81.94%	51.19-52.82%	75.45–81.29%
classification accuracy			63–70%	

**Table 3 animals-15-01984-t003:** Measurement equipment and signal properties.

Detected Signal	Equipment Photos	Type	Brand	Sampling Rates (Hz)	Measurement Frequency (Hz)	Signal-to-Noise Ratio (dB)
SP	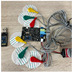	STM32	ST	1000	10–110	-
MP	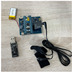	STM32	ST	1000	20–500	-
RF	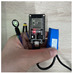	ESP32	Espressif Technologies	20	50–500	-
VP	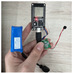	ESP32	Espressif Technologies	8000	0–3400	60–65

**Table 4 animals-15-01984-t004:** The division of the training set and the test set.

Algorithms	Number of Subjects	Sample Happiness Mood	Sample Sadness Mood	Sample Anger Mood	Sample Fear Mood	Sum
Training Set	22	141	87	72	79	379
Test Set	8	46	28	31	39	144
Sum	30	187	115	103	118	523

**Table 5 animals-15-01984-t005:** The 29 signal features extracted for each signal.

Domain	Feature Name Abbreviation
time domain	q1,q3, median, mean, std, var, rms
	min_ratio, max_ratio
	diff1_mean, diff1_median, diff1_std
	diff2_mean, diff2_median, diff2_std
frequency domain	mean_f, median_f, std_f, var_f, rms_f
	min_ratio_f, max_ratio_f
	diff1_mean_f, diff1_median_f, diff1_std_f
	diff2_mean_f, diff2_median_f, diff2_std_f
nonlinear	mcr

**Table 6 animals-15-01984-t006:** Part of the experimental data.

Sample Num	SP (μS)	MP (μV)	RF (Times)	VP (Hz)	Emotion
1	8.3	18.5	57	1217	Happiness
2	6.4	15.2	46	1091	Happiness
3	5.8	14.6	39	927	Sadness
4	6.7	16.1	42	818	Sadness
5	25.1	65.8	71	423	Anger
6	19.2	61.9	74	228	Anger
7	13.9	42.2	65	1609	Fear
8	12.4	49.3	83	1940	Fear

**Table 7 animals-15-01984-t007:** SP signals of four breeds of dogs under four emotions. (Abscissa Label: SP signal frequency (Hz) Ordinate Label: Time (s)).

	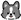 Border Collie	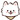 Samoyed	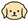 Golden Retriever	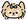 Shih Tzu
Happiness	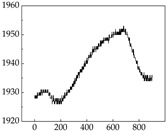	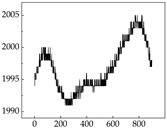	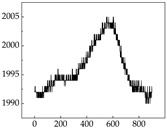	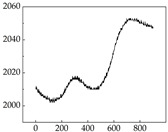
Sadness	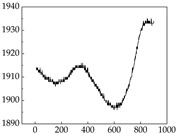	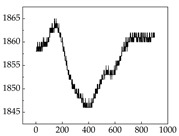	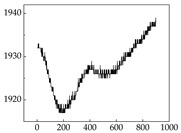	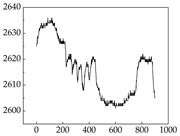
Anger	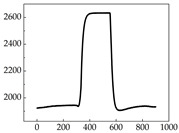	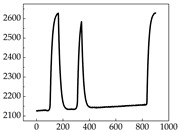	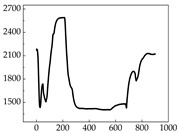	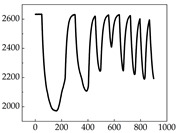
Fear	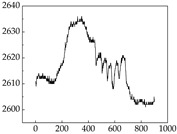	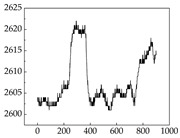	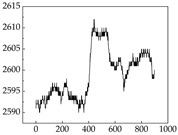	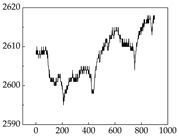

**Table 8 animals-15-01984-t008:** MP signals of four breeds of dogs under four emotions. (Abscissa Label: MP signal frequency (Hz) Ordinate Label: Time (s)).

	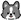 Border Collie	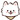 Samoyed	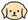 Golden Retriever	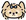 Shih Tzu
Happiness	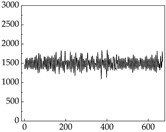	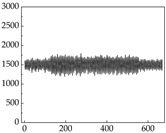	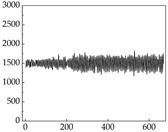	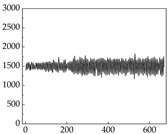
Sadness	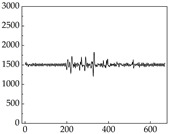	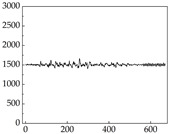	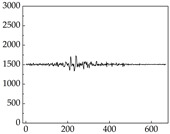	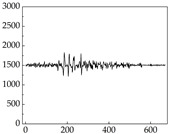
Anger	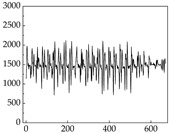	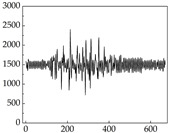	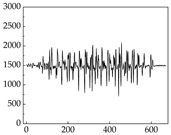	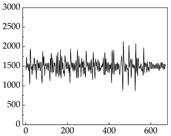
Fear	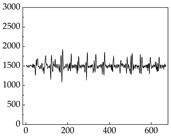	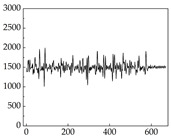	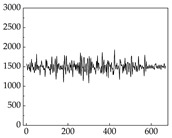	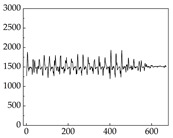

**Table 9 animals-15-01984-t009:** The hyperparameter settings for the NN [28] classifier model.

Hyperparameters Symbol	Value	Explanation
hidden_layer_sizes	(50,100,100,50,40)	hidden layer structure
solver	‘lbfgs’	weight optimization methods
alpha	1 × $10^{-5}$	regularization parameter

**Table 10 animals-15-01984-t010:** The hyperparameter settings for the SVM [29,30] classifier model.

Hyperparameters Symbol	Value	Explanation
C	17	penalty coefficient
kernel	‘rbf’	kernel function
gamma	0.001	kernel function coefficient

**Table 11 animals-15-01984-t011:** The hyperparameter settings for the GBDT [25,31] classifier model.

Hyperparameters Symbol	Value	Explanation
n_estimators	100	maximum number of iterations for weak learners
max_depth	3	maximum depth of the decision tree
min_samples_split	2	minimum number of samples contained in each non-leaf node
min_samples_leaf	1	minimum number of samples contained in each leaf node

**Table 12 animals-15-01984-t012:** The hyperparameter settings for the **XGBoost(PRO)** classifier model.

Hyperparameters Symbol	Value	Explanation
learning_rate	0.1	learning rate of the model
max_depth	6	maximum depth of the decision tree
gamma	1	minimum loss reduction required for node splitting
objective	‘multi:softprob’	define task types
num_class	4	number of defined categories
min_child_weight	1	minimum value of the sum of leaf node sample weights
subsample	0.8	proportion of samples when training each tree
n_estimators	90	maximum number of iterations for weak learners

**Table 13 animals-15-01984-t013:** Comparison of classification reports for NN, SVM, GBDT, **PRO**.

	Algorithmic	AUC	Accuracy	Recall	F1 Score	Support Number
	NN	0.78	0.86	0.72	0.70	326
Training	SVM	0.82	0.87	0.78	0.76	330
Set	GBDT	0.89	0.91	0.84	0.83	345
	**PRO **	**0.91 **	**0.92 **	**0.88 **	**0.88 **	**347 **
	NN	0.75	0.83	0.69	0.71	119
Test Set	SVM	0.76	0.85	0.72	0.74	122
	GBDT	0.83	0.88	0.81	0.77	127
	**PRO **	**0.85 **	**0.90 **	**0.87 **	**0.79 **	**130 **

## Data Availability

The original contributions presented in this study are included in this article. Further inquiries can be directed to the corresponding author.

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
