# Peer review of "A Multi-Epiphysiological Indicator Dog Emotion Classification System Integrating Skin and Muscle Potential Signals"

_animals, 2025, doi:10.3390/ani15131984_

Round 1
Reviewer 1 Report
Comments and Suggestions for Authors
The study presents a valuable and innovative approach to canine emotion classification using physical signals and machine learning. The functioning is sound, and the results are promising. Some explanations and improvements will be extended to the clarity and perfection of the manuscript, mainly about vocabulary, figures and experimental details. Please see the specific comments given below.
- Ensure that all full meanings (eg, SP, MP, RF, RF, VP, Shape, SVM, etc.) are written on the first mention and continuously used throughout the manuscript.
- Provide more information about signal acquisition hardware- Sampling rate, signal preprocessing phase, and duration of each emotional state recording
- Some figures lack a detailed caption or axis label (eg, figures 5-7). Increase clarity by describing what is shown in each figure caption, including units and references.
- Add a detailed description of the labeling of emotional stages (happiness, sadness, fear, anger). Consider behavioral signal, expert verification, or video coding references to support the accuracy of emotion classification
- Expand the ethics section to explicitly mention how fear and sadness induction procedures ensured animal welfare (e.g., signs of stress, use of enrichment afterward, withdrawal protocols).
Reviewer 2 Report
Comments and Suggestions for Authors
This is an interesting paper with a well-thought out research design, measuring dog emotions using for different sensor. The paper is well-readable, however the English still needs to be improved.
There are also other areas of improvement:
- there should be a list of all abbreviations, or at least the abbreviations should be spelled out and mentioned the first time they are used in the paper.
- the induction of the dog emotions could be described in a more systematic and structured way
- the results could be presented more clearly. Do I understand correctly that you have 4 times 29 features, then it is not surprising the XGBoost works best, as your N is somewhat small.
- It would be interesting to know which features predict which mood states best.
- One problem is the definition of ground truth. this is related to 2, as the reader does not know how good the owners are in accurately predicting the emotions. It would be advisable to have these assessments checked by an expert.
paper is well readable, but there are small glitches throughout the paper
Reviewer 3 Report
Comments and Suggestions for Authors
This manuscript introduces skin potential (SP) and muscle potential (MP), combined with traditional respiratory frequency and vocal patterns, which offers practical value for canine emotion recognition and its real-world application. However, there are several major issues that need to be addressed before publication:
The authors do not provide any visual illustrations of how emotion labels were assigned, nor do they show photographs or behavior diagrams of dogs in each emotional state. This lack of visual evidence undermines the credibility of the emotion labeling process.
The types, brands, sampling rates, measurement frequency, signal-to-noise ratio, and data quality control strategies of the sensors are not clearly described.
It is unclear whether the accuracy differences between XGBoost and other models (e.g., GBDT) are statistically significant, and whether an independent validation set was used is not specified.
The "contribution rate" shown in Figure 6 lacks explanation on how it was calculated, what the unit of measurement is, and whether feature interactions were considered.
Reviewer 4 Report
Comments and Suggestions for Authors
Overall: The study is interesting and novel and can help improve animal welfare and wellbeing under human handling conditions. There are some additional pieces of information that are needed as well as some parts that need to be clarified or referenced.
Abstract: Overall the abstract is good and covers the purpose, methods, findings, and application. It would benefit from full disclosure of how many breeds and animals were part of the dataset. Also, line 11 – please clarify what “features” indicates (..most influential features of emotions, for example).
Introduction:
Overall the introduction is good. Please see below for additional comments.
Line 21: Abnormal emotion – all emotions are valid. It might be beneficial to clarify this as negative emotional affect or specifically anger, fear, rage, etc.
Table 1 should have a direct citation as to where these indicators originated. If it originated from multiple sources, then add another column to indicate where each behavioral signal originates in the literature.
Line 61: Nake the reference that is currently written as “Ref.”
Same for Line 64
Lines 74-79: This description of methodology should be in the methods section.
Line 87: Again, classify “abnormal”
Line 94: “raising” should be changed to encompass handling, training, and management.
Materials and Methods:
The materials and methods need additional information to clarify how the study was conducted and who was responsible as well as where information was obtained.
Lines 108-109: There are claims of different environments relating to emotional control in animals. This needs to have a reference and citation to support the use of these spaces.
It also needs to be clear who was judging the dog’s emotions based on behavior and how they were trained to determine this. Owners often misunderstand their animals and do not always accurately assess emotional expression of dogs. It’s important to clarify who was responsible for classifying the emotions and how they were trained.
Also, how was the dog trained to the device? How was the dog desensitized to wearing the device? How were reactions to the device and equipment accounted for in behavioral expressions?
How many researchers were coding? Was it done live or through pictures or videos? Was their interrater reliability?
Results:
Overall, the results are good. Please see below for comments.
Line 225-226: There is a claim that small dogs are sensitive. This needs to have a citation/reference for this claim.
The term “ups and downs” should be changed to fluctuations in emotional expression or affect.
It would also be good to emphasize again HOW the fluctuations were matched with existing methods of reading/evaluating/measuring emotions (ie ethograms and behavioral indicators from research)
Lines 255-256: The claims around emotions and skin secretions need to have a citation/reference.
Again, the term “abnormal emotions” needs to be clarified and anchored in literature on animal emotions.
Discussion and Conclusion:
Overall, the discussion and conclusion are thorough and help contextualize the findings
Round 2
Reviewer 2 Report
Comments and Suggestions for Authors
all my concerns have been addressed.
Comments on the Quality of English Language
The English has some very small glitches. (a few missing pronouns)